# From deterministic to fuzzy decision-making in artificial cells

Ferdinand Greiss [1], Shirley S. Daube [1], Vincent Noireaux [2] & Roy Bar-Ziv [1]✉

Building autonomous artificial cells capable of homeostasis requires regulatory networks to gather information and make decisions that take time and cost energy. Decisions based on few molecules may be inaccurate but are cheap and fast. Realizing decision-making with a few molecules in artificial cells has remained a challenge. Here, we show decision-making by a bistable gene network in artificial cells with constant protein turnover. Reducing the number of gene copies from $10^5$ to about 10 per cell revealed a transition from deterministic and slow decision-making to a fuzzy and rapid regime dominated by small-number fluctuations. Gene regulation was observed at lower DNA and protein concentrations than necessary in equilibrium, suggesting rate enhancement by co-expressional localization. The high-copy regime was characterized by a sharp transition and hysteresis, whereas the low-copy limit showed strong fluctuations, state switching, and cellular individuality across the decision-making point. Our results demonstrate information processing with low-power consumption inside artificial cells.

[1] Department of Chemical and Biological Physics, Weizmann Institute of Science, Rehovot 76100, Israel. [2] Department of Physics, University of Minnesota, Minneapolis, MN 55455, USA. ✉email: roy.bar-ziv@weizmann.ac.il

Making well-informed decisions takes time and energy. The fundamental tradeoff between those three aspects is exemplified by recognition memory in cognitive tasks[1], optimality in molecular recognition as an error-correction mechanism[2,3], and cellular adaptation in chemo-sensing[4]. Decisions on a cellular level are made by regulatory proteins that integrate information from the environment and elicit a response by modulating RNA and protein production. In each cell, the copy number of regulatory molecules could vary between one to a few hundred, subjecting the information processing to random fluctuations[5,6]. The fluctuations can stem from production and degradation of proteins, and the binding and unbinding of proteins to and from their regulatory sites on DNA. Theoretical considerations suggest that a decision becomes exponentially more stable as the copy number of regulatory elements, hence metabolic load, increases[7,8].

Decision-making of a genetic regulatory network (GRN) should be precise and deterministic when the process is averaged over many molecules. In the small-number limit, fluctuations in gene expression with a few molecules, hence little averaging, may reduce the precision and lead to fuzzy decision-making dominated by fluctuations (Fig. 1a, b). In this work, by measuring the decision-making characteristics of a GRN, both in the high and low gene density regimes, we find that the decision-making becomes fuzzy at low gene densities due to noise in the binding and unbinding of transcription factors. We demonstrate that gene regulation in artificial cells appears at lower concentrations than necessary for the equilibrium binding of transcription factor to its target sites, and that the decision-making of the GRN trades speed with accuracy through the transition from low to high gene densities.

## Results

### A minimal bistable genetic network in an artificial cell.
In order to experimentally test the properties of biological decision-making

without background reactions such as DNA replication, genetic cross-talk, resource sharing, or cell size variations[9–11], we used a programmable cell-free system of gene-expression reactions with constant protein turnover[12]. We built artificial cells defined by a compartment with a 20 μm radius and 3.5 μm height, connected by a thin capillary ($L = 90\,\mu m$ and $W = 7\,\mu m$) to a reservoir of cell-free extract to support transcription-translation (TXTL) reactions (Fig. 1c, Supplementary Fig. 1)[13–15]. The compartment volume was $3.8 \times 10^3\,\mu m^3$, roughly 1000-fold larger than a typical bacterial cell. The capillary allowed free diffusion in and out of the compartment, creating TXTL dynamics with an apparent protein lifetime of $\tau = \pi R^2 L / WD = 8\,min$[13], with $D = 30\,\mu m^2\,s^{-1}$ as a typical protein diffusion coefficient[13]. We used an elastomer to create the compartments, bonded to a coverslip to detect the synthesis of single fluorescent reporter proteins from the DNA attached to the glass substrate (see the "Methods" section).

We based our genetic model system on a bistable decision-making GRN built from the core elements of the lambda bacteriophage regulatory network (Fig. 1b)[16–20]. In its native environment, this GRN controls the phage's decision to lyse the cell or integrate its genome into the bacterial host genome thus entering the lysogenic state. This state can be maintained for more than $10^5$ generations by the presence of only a few hundred copies of proteins before switching back to the lytic phase[9,16,19,21,22]. The minimal GRN consists of two transcription factors, CI and Cro repressors, which mutually inhibit each other's production by binding to their respective promoters ($P_R$ and $P_{RM}$) (Fig. 1b)[16]. By analogy to digital electronics, an ideal bistable GRN can be viewed as a latch circuit that activates either of two promoters and remembers the active promoter until toggled (Fig. 1b). CI has been shown to be the main regulator, responsible for entering and maintaining the lysogenic state, while Cro buildup serves a tipping point to decisively and irreversibly enter the active $P_R$ promoter (lytic) state[23]. This

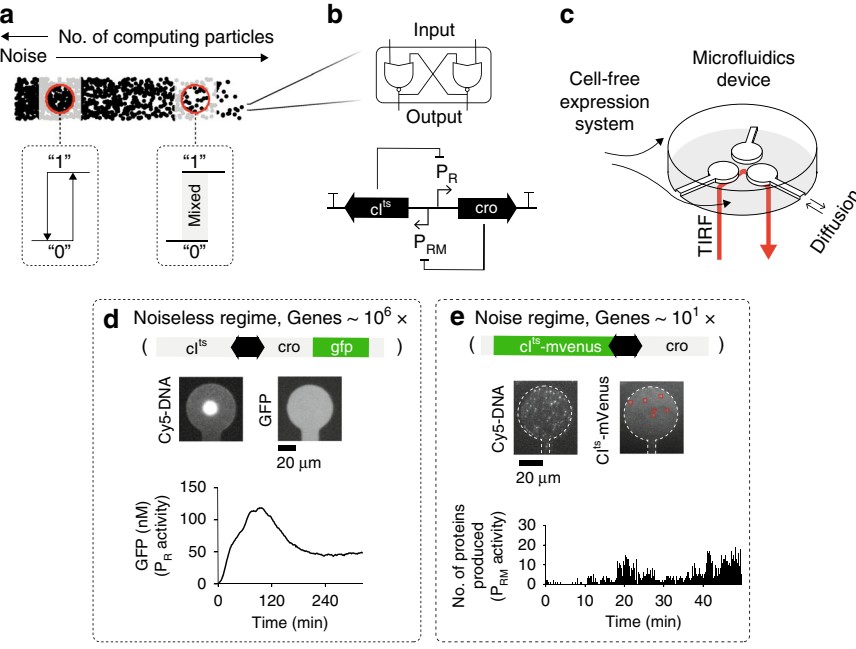

**Fig. 1 Biological decision-making in an artificial cell at high and low gene densities. a** Number of particles influence decision-making by an effective noise level: a bistable GRN responds to an input with a binary output in the noiseless regime at large numbers, or by a fuzzy output in the noise regime at small numbers. **b** Scheme of the minimal bistable GRN based on CI and Cro transcription factors repressing each other's promoter with analogy to an electronic latch circuit. **c** Scheme of a microfluidic device with three compartments as artificial cells and total internal reflection fluorescence (TIRF) microscopy to monitor protein production at low gene densities in a cell-free expression system[30]. **d** Cy5-DNA assembled in a compartment at high density gives rise to GFP expression reporting on the $P_R$ activity. **e** Cy5-DNA at single-molecule resolution inside the compartment. Single CI[ts]-mVenus were integrated over the compartment to give the number of proteins produced in 15 s, reporting on the overall $P_{RM}$ activity. Source data are provided as a Source data file.

inherent asymmetry between CI and Cro is due to the promoters' architecture including auto-inhibition and auto-activation loops[16]. To toggle the promoter activities, we used a temperature-sensitive CI (CI[ts]) mutant that tunes its deactivation rate with a rise in temperature from 30 °C (no deactivation) to 41 °C (fast deactivation)[18,24].

We immobilized the DNA constructs with the bistable GRN on the surface of the compartments. At the high-density regime, we packed roughly $10^5-10^6$ copies[25] in a DNA brush patterned on a circle of 14 μm diameter ("Methods"). We placed the gene coding for green fluorescent protein (GFP) in tandem with the *cro* gene, as a reporter of the activity of the strong $P_R$ promoter[16,18] (Fig. 1d), and recorded the signal using epi-fluorescence microscopy. The strong $P_R$ promoter increased the GFP signal over the background fluorescence of the elastomer used to create the compartments. We observed smooth GFP dynamics with a high signal to noise ratio, peaking after ~2 h and slowly reaching steady-state values after two more hours (Fig. 1d).

For the low-copy regime, we immobilized a small number of DNA copies with an average of ~20 DNA molecules of the bistable GRN in each compartment, which amounted to a very low effective DNA concentration of ~10 pM (Fig. 1e and Supplementary Fig. 2). We could not further reduce the number of DNA molecules while maintaining a narrow distribution due to the diffusion-based stochastic immobilization in our artificial cells. To allow the precise monitoring of protein production[26], we directly fused the *cI[ts]* gene to the fast-maturing fluorescent protein mVenus[27] under the weak $P_{RM}$ promoter[16,18], and circularized the DNA constructs to minimize degradation by exonucleases in the *E. coli* extract (see "Methods", Supplementary Fig. 3, Supplementary Table 1). We employed total internal

reflection fluorescence (TIRF) microscopy with a penetration depth comparable to the typical dimension of DNA molecules to observe localization of the proteins close to the coverslip surface.

After introducing the cell-free extract, we could observe the appearance of discrete CI[ts]-mVenus fluorescent spots that were single-peaked and rapidly bleached with a lifetime of ~4 s (Supplementary Fig. 4). The fast bleaching removed all newly produced proteins that localized to the surface. We could not reliably detect repeated binding of proteins to specific locations due to nonspecific adsorption on the surface upon heat-inactivation of CI[ts]-mVenus (Supplementary Fig. 5). Therefore, we integrated the number of produced proteins inside the compartment independently of the location into 15 s time intervals, reporting on the overall $P_{RM}$ activity of the artificial cell (Fig. 1e)[18,28,29]. In sharp contrast to the high-density regime, we observed strong fluctuations of protein production rates that reached a first maximum after 20 min.

**Decision-making in an artificial cell at low and high gene densities.** Because we optimized the cell extract such that the basal protein production rate varied by no more than ~10% for 31–41 °C (Supplementary Fig. 6), any change in signal properties as a function of temperature could be attributed to a response of the GRN to a change in CI[ts] deactivation rate. Based on the GRN architecture studied in bacteria[9,16,18], we anticipated initial low $P_{RM}$ and low $P_R$ activities at low temperature, where stable CI[ts] first represses $P_R$ and then self-represses $P_{RM}$ at the mutual promoter sites (Fig. 2a, state 1). An increase in CI[ts] deactivation rate at higher temperature should reduce the occupancy of CI[ts] at the promoters, leading to self-activation of CI[ts] and gradual

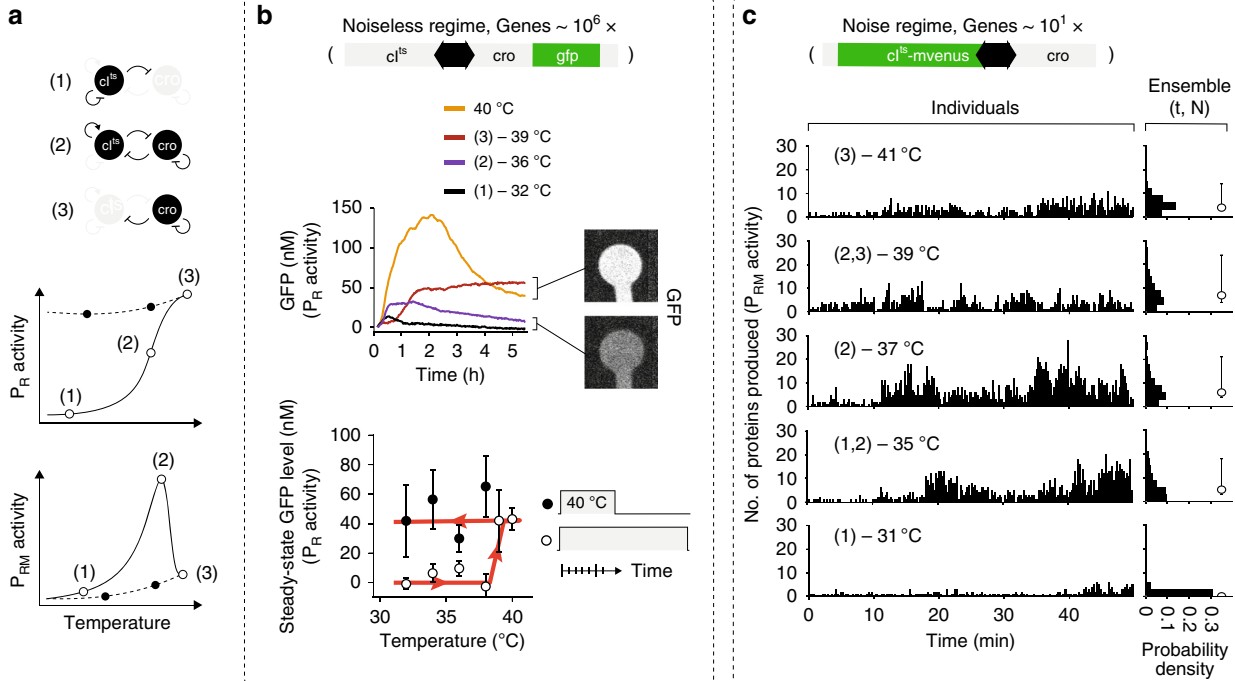

**Fig. 2 Bistable GRN and decision-making at high and low gene densities. a** Illustration for the expected output of $P_R$ and $P_{RM}$ activity as a function of temperature, based on the known molecular feedbacks of the GRN. **b** Dynamics of GFP expression reporting on the $P_R$ activity at high DNA density (upper panel). Solid lines depict average dynamics, $N = 24$. (1), (2), and (3) relate to molecular states of the bistable GRN at each temperature, as in panel a. Steady-state GFP levels averaged over individual compartments (lower panel). Temperature was either kept constant throughout the measurement (open circle) or after a 40 °C initial incubation step (closed circle). Error bars show the mean and standard deviation (SD) of compartments. Red line drawn to guide the eye. **c** Number of CI[ts]-mVenus produced in 15 s to give production rates (left column) and ensemble distribution over time $t$ and compartments $N$ after >20 min (right column) in the low-density regime. Circle with error bars give the median and 32th to 68th percentile of the ensemble production rates ($N = 1044, 696, 1044, 2088, 1044$ in the order of increasing temperature). Source data are provided as a Source data file.

production of Cro from $P_R$ (Fig. 2a, state 2). At higher temperatures, complete deactivation of $CI^{ts}$ leads to fully active $P_R$ (Fig. 2a, state 3).

At the high-density regime with GFP, hence Cro, reporting on $P_R$ activity, no signal was recorded at 31 °C, indicating an active $P_{RM}$ promoter (Fig. 2b, upper panel, Supplementary Fig. 7, and Fig. 2a, state 1). We observed a decisive transition between 38 and 39 °C to high GFP steady-state levels, switching to active $P_R$ promoter (Fig. 2a, state 3). To check the stability of the activated $P_R$ promoter, we first expressed the proteins in the monostable region at 40 °C for 1 h, followed by a drop to lower temperatures. $P_R$ activity was reliably maintained (Fig. 2b, lower panel and Supplementary Fig. 7), independently of the compartment geometry and hence protein lifetime (Supplementary Fig. 8). Control experiments with a GRN lacking the *cro* gene (a monostable GRN) resulted in a similar initial response to temperature variations, but with overall lower GFP levels due to the lack of commitment to the $P_R$ promoter by Cro, as was already observed in bacteria (Supplementary Fig. 9)[23], suggesting that Cro was required to obtain decisive switching (Fig. 2b, lower panel). Overall we observed a deterministic response to the temperature input, most likely due to low noise levels, with precise but slow dynamics on a characteristic timescale of ~1 h (Supplementary Fig. 10), consistent with the bistable feature of the GRN as an ideal memory device. Solution experiments reconstituted a similar temperature response of the different GRNs, but could not identify a decisive switching temperature since the closed system accumulated proteins without turnover, and hence did not reach steady-state dynamics (Supplementary Fig. 11).

In the low-copy regime, with $CI^{ts}$-mVenus reporting on the $P_{RM}$ activity, the overall production rates of an ensemble of compartments (Supplementary Fig. 12, Fig. 2c, right column and Supplementary Table 2) increased (Fig. 2a, state 2) and decreased again at higher temperature as anticipated (Fig. 2c and Fig. 2a, state 3). We observed variability between compartments only in the 35–39 °C range as computed from the standard deviation of time-averaged production rates (Supplementary Fig. 13). In this range, production rates on the time scale of many minutes (~5 min) waned and grew again (Fig. 2c, Supplementary Figs. 10 and 12). In the absence of Cro, we expected to observe a $P_{RM}$ activity without inhibition by Cro, namely, no display of spontaneous transitions between the two promoters. Indeed, the temperature response for the monostable GRN was similar to the bistable GRN (Figs. 2c and 3a, b, respectively, and Supplementary Fig. 12), but occurred across a narrower temperature range with higher median production rate at 37 °C (Figs. 2c, 3a, and Supplementary Table 2), and reduced variability at 37 and 39 °C (Supplementary Fig. 13). A control experiment of a monostable GRN with the wild-type *cI* gene displayed almost no temperature response, consistent with the temperature having no effects on the GRN other than the deactivation of $CI^{ts}$-mVenus (Supplementary Fig. 14 and Supplementary Table 2). To exclude the possibility that the high concentration of the TXTL machinery in the cell-free extract[30] governed the GRN dynamics, we replaced the $P_{RM}$ by a consensus promoter sequence and added a ribosomal binding site (RBS) to the natural leaderless mRNA (Supplementary Fig. 15)[31]. We observed a strong increase in production rates in both cases. These control experiments supported $P_{RM}$ activity as the rate-limiting step of the GRN, also validating CI as the main regulator of $P_{RM}$. We also found no dependence of protein production rates on the slightly varying number of DNA molecules in the low-density compartments (Supplementary Fig. 1). The data therefore suggests that in the

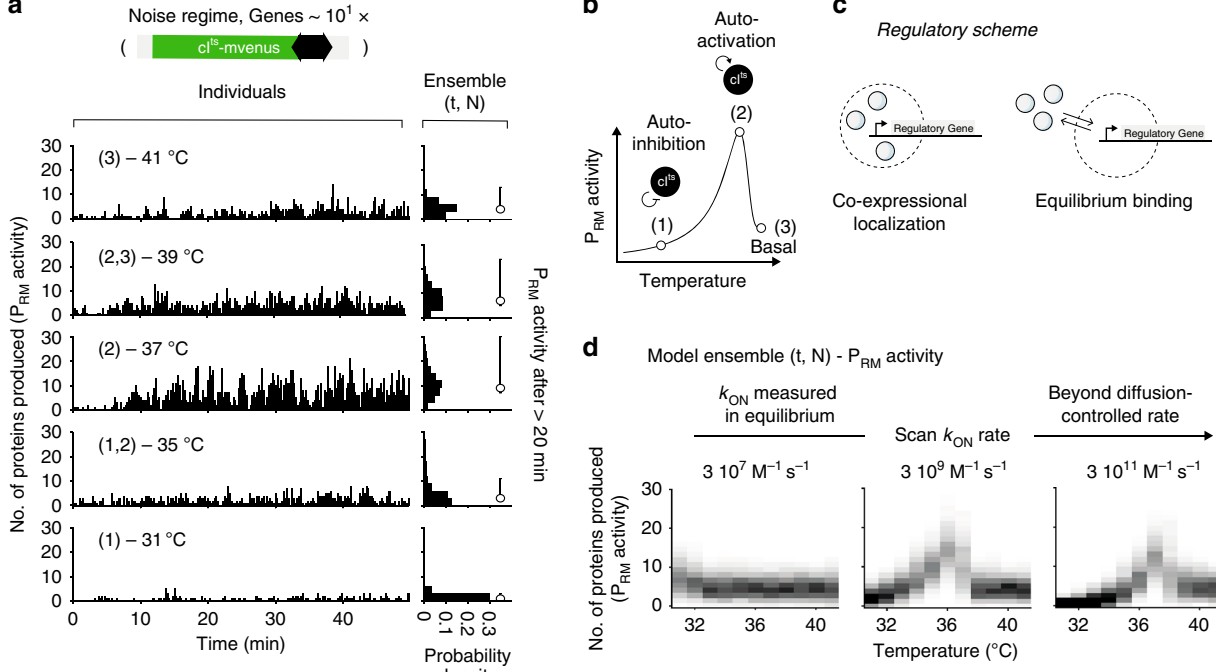

**Fig. 3 Monostable GRN and co-expressional localization at the low-density limit. a** Production rate of $CI^{ts}$-mVenus proteins in 15 s inside individual compartments (left column) and ensemble production rates (right column) as in Fig. 2c. Circle with error bars give the median and the 32th to 68th percentile of the ensemble production rates ($N$ = 1044, 1392, 1044, 1044, 1044 in the order of increasing temperature). **b** Illustration of the expected $P_{RM}$ activity with temperature in the monostable GRN and the corresponding states. **c** A protein repressor synthesized in proximity to its DNA-binding site, reach higher effective concentrations by co-expression localization (left scheme) than in an equilibrium scenario due to diffusion away from the DNA (right scheme). **d** Ensemble production rates obtained from stochastic simulations with a scan in the $CI^{ts}$ binding rate to the DNA-binding sites. Source data are provided as a Source data file.

bistable GRN either one of the two promoters was randomly activated and could spontaneously transition to the other promoter, but only at the 35–39 °C temperature range[18]. We also note that the variability occurred in a broad temperature range close to the switching point of the high-density regime.

**Gene regulation at low-copy numbers suggests co-expressional localization.** Whereas the long time (~1 h) to reach a final decision seemed to come with high precision at an energetic cost of ~$10^6$ proteins in the high-density regime, fuzzy decisions appeared with an increase in speed and only a handful of proteins in the low-density regime (Supplementary Fig. 10). Still, we wondered how decisions could be realized in the low-density regime considering the low DNA and protein concentration. We estimated the total number of proteins produced in 50 min to be ~1000, which amounted to an upper-limit concentration of ~50 pM. With DNA at 10 pM, the concentrations were below the measured CI binding affinities of ~3 nM[32] and ~50 nM[11], in vitro and in vivo, respectively, suggesting that equilibrium binding considerations could not account for the binding of nascent CI[ts] to its DNA-binding site. Considering the typical binding rate of CI to its operator sites of $k_{on} \approx 3 \times 10^7$ M$^{-1}$ s$^{-1}$ [33], it would take a single protein much more than 20 min to bind and modulate production rates, slower than the observed time to reach steady states. These considerations could be reconciled by a scenario in which regulatory proteins are localized to the DNA during production and kept close to their operator sites with an increase in local effective concentration (Fig. 3c). Since our system did not allow us to observe promoter activity, mRNA production, and protein synthesis in parallel, we corroborated this notion by stochastic simulations ("Methods") and found that they reproduced the experimental results only at values 100-fold higher than previously measured CI $k_{on}$ rates in equilibrium (compare Fig. 3a

to Fig. 3d, Supplementary Table 3, and Supplementary Fig. 16). Likewise, CI[ts] repressed the production of GFP in solution experiments more efficiently when produced on the same plasmid than on a separate plasmid (Supplementary Fig. 17). This notion of spatial coupling between protein synthesis and DNA binding as an enhancement mechanism for site location was also discussed in a bacterial context[34].

**Fluctuations across the decision-making transition.** Next, we sought to find the source for the fuzzy decision-making in the low-density regime. The Fano factor (variance/mean) was computed as a measure of fluctuations in CI[ts]-mVenus production rates in single compartments after 20 min and plotted in Fig. 4a. The Fano factor equals 1 for a random Poissonian process and deviates from 1 when fluctuations in the gene-expression dynamics increase. We found its value to be close to 1 and constant with temperature for the wild-type monostable GRN. For both mono- and bistable GRNs, the Fano factor was >1 around 37 °C, and correlated with the time-averaged production rates (Fig. 4a, upper row, Supplementary Fig. 18). In addition, the Fano factor of the bistable GRN had a larger spread between 35 and 39 °C in the range of large variability. The data indicated that the regulation of the $P_{RM}$ activity by CI[ts]-mVenus strongly fluctuated only around 37 °C. Here, the fast inactivation and therefore lower occupancy of CI[ts]-mVenus at the $P_R$ promoter seemed to reduce the stability of the auto-activation of $P_{RM}$ activity by CI[ts]-mVenus (Fig. 4b and Supplementary Fig. 19); hence a lower occupancy at the $P_R$ promoter allows the leaky production of Cro to induce the observed variability and spontaneous transitions between promoters in the bistable GRN, resulting in fuzzy decision-making.

To extract typical correlation time scales in the fluctuating production rates of CI[ts]-mVenus, we computed the autocorrelation

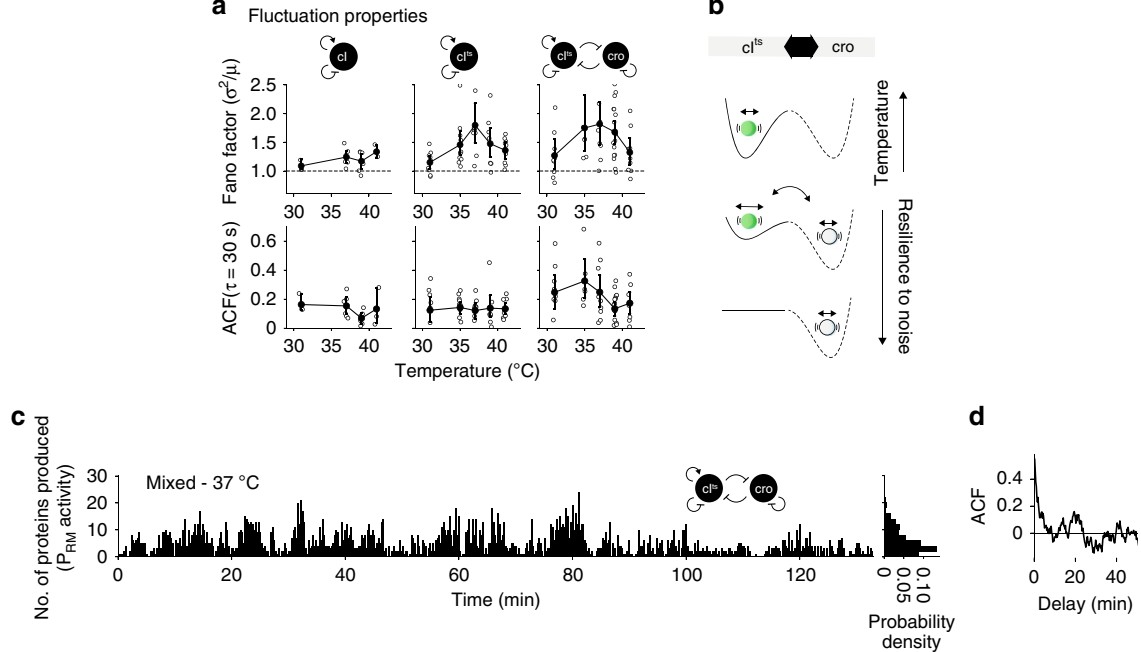

**Fig. 4 Fluctuations and spontaneous transitions at the low-density limit. a** Fano factor (variance over mean) with Poissonian process (dashed line) for the GRNs at various temperatures (upper row). Amplitude of temporal autocorrelation function (ACF) of fluctuating production rates (as shown in Figs. 2c, 3a) in individual compartments at $\tau = 30$ s. Error bars were bootstrapped and show mean and SD of compartments (see Supplementary Fig. 12 for the bistable and monostable GRN and Supplementary Fig. 14 for the wild-type monostable GRN). **b** Scheme of noise, active promoter transitions, and memory in the bistable GRN. Short-term fluctuations in gene expression (circle within a well) originate from local noise. Long-term fluctuations can originate from spontaneous promoter transitions with the deactivation of CI[ts]. **c** Number of proteins produced in 15 s for long-term experiment of the bistable GRN at 37 °C. **d** The ACF of the trace in panel c. Source data are provided as a Source data file.

functions (ACF) in each compartment after 20 min in steady-state, which averages over the fluctuations and shows the correlation of a signal between time points. The ACF amplitude at a delay of $\tau = 30\,s$ had a higher amplitude for the bistable GRN than the wild-type and monostable GRN at 39 °C and below (Fig. 4a, lower row and Supplementary Fig. 18). This may indicate more correlated short-term fluctuations of CI$^{ts}$-mVenus production rates for the bistable GRN as was observed for a close bifurcation point in other complex dynamic systems, and can be explained by a reduced recovery rate from perturbations[35]. Here, molecular perturbations may stem from the leaky production of Cro from $P_R$ within the region of bistability (Fig. 4b). To get better statistics on the long-term fluctuations that could last for many minutes in the bistable GRN (Fig. 2c), we extended the duration of the experiment to 130 min at 37 °C (Fig. 4c). We observed highly fluctuating production rates with repeating signatures of protein bursts, each lasting for several minutes (Fig. 4d and Supplementary Fig. 20). A strongly damped periodic signal could be observed in individual ACFs with a period of ~10 min, but averaged out when ensemble-averaged (Supplementary Fig. 20).

## Discussion

In conclusion, we established a minimal decision-making GRN, with and without variability and spontaneous transitions, in an artificial cell devoid of any background reaction in living cells. Dynamics could be monitored at the low and high-density limits of genetic decision-making circuits, demonstrating a clear tradeoff between slow and precise versus fast and fuzzy. We suggest co-expressional localization to enhance binding rates (Fig. 3)[34], a nonequilibrium mechanism that seems to be essential for realizing gene regulation with a few molecules in our artificial cells.

The timing of gene expression in artificial cells based on GRNs will depend on the copy number of molecules. We discuss the difference in the observed timescales required for decision-making in the two DNA density regimes. Without the turnover of mRNAs and proteins, a compartment will be quickly filled with the molecules of a single outcome that prevent the GRN from reversing its decision. The production is therefore balanced by the removal (degradation, dilution, and deactivation) to reach a steady-state copy number that can give an estimate of how long a decision takes. For the high-copy number regime, the removal of many molecules down to the threshold value where the GRN can flip the switch takes longer. Hence, the speed of decision-making reduces with a larger steady-state copy number, while the accuracy increases due to the averaging over many molecules. Assuming a simple model with the removal of molecules as a first-order reaction and a copy number threshold to flip the GRN, we can compute a time difference of $\Delta t = \tau \ln R$ with $R$ as the ratio of steady-state copy numbers. Together with a protein lifetime of ~8 min in an artificial cell and the ratio of 1000 as we approximately probed with the low- and high-density regime (Fig. 1d, e), we obtain a ~55 min time difference in the decision-making. Despite the simplicity of the model, we can reproduce the calculation with an experimental decision-making time of ~5 min and ~1 h in the low- and high-density regime, respectively (Fig. 2b, c and Supplementary Fig. 10). Our findings in isolated artificial cells may also shed light on the tolerance of living cells for fuzzy, but timely decision-making.

Finally, we discuss the proposed mechanism of co-expressional localization in light of the apparent enhancement of rates suggested in bacteria. Mechanisms for DNA target search range from the 1D sliding of transcription factors along DNA[36–39], 3D hopping of transcription factors between nearby DNA segments[36–39], slow dispersion of mRNA and proteins by partitioning the cell[40], and the colocalization of transcription factor

genes with their target regulatory sites on the genome[34]. These mechanisms provide a solution to the search problem of transcription factors for the highly compacted chromosome and a large number of unspecific DNA-binding sites inside living cells. Due to the small amount of DNA and proteins (~pM) below the equilibrium binding in the low-density regime in our system, the stronger repression of GFP when the $cI$ gene is localized together with the $gfp$ gene, and the lack of expected CI-mVenus clusters from partitioning in cell-free expression systems[41,42], we suggest co-expressional localization as mechanism for the gene regulation in the artificial cells.

Building life-like cells is a major step toward understanding life itself[43–45]. Our results demonstrate biological decision-making in artificial cells. In combination with the detailed knowledge on natural gene regulation, the de novo construction of synthetic GRNs[46], and metabolic pathways[47,48], artificial cells may lead to autonomous agents operating at the single-molecule level with low-power consumption.

## Methods

**Cell-free transcription-translation.** Cell-free expression was carried out using an *E. coli* TXTL system as previously described[30,49]. Briefly, *E. coli* cells (BL21, Rosetta2, Novagen) are grown in 2xYT medium supplemented with phosphates. Grown cells (OD600 = 1.5–2) were lysed either by bead beating or with a cell-press. After centrifugation (12,000 × g for 10 min), the supernatant was recovered and preincubation at 37 °C for 80 min. After a second centrifugation step (12,000 × g for 10 min), the supernatant was dialyzed for 3 h at 4 °C. After a final spin-down (12,000 × g for 10 min), the supernatant was aliquoted and stored at −80 °C. The cell extract (29 μL) was thawed on ice, supplemented with the necessary solutions (10 mM Mg-glutamate, 80 mM K-glutamate, 4% PEG 8000, 10.8 mg/ml Maltodextrin, amino acids, energy regeneration, and GamS), filled to 87 μL with water, and gently mixed.

**Reagents.** All primers were ordered from IDT. All PCRs were performed with the KAPA HiFi HotStart ReadyMix (Kapa Biosystems, Roche). All PCR products were purified with Promega Wizard SV-Gel and PCR Clean-Up. All plasmids were derived from the pBEST backbone[30] with Ampicillin selection marker and ColE1 origin of replication. Cloning was performed with a *E. coli* strain (DH5α) and plasmids were extracted with spin column purification (Promega).

**Assembly of GRNs.** The GRNs with $cI^{ts}$, $O_R3$, $O_R2$, $O_R1$, and $cro$ were directly isolated from lambda phage DNA (methylated lambda phage DNA, Sigma) using primers P3 and P4 for transfer-PCR into the pBEST vector to obtain a plasmid with the minimal bistable GRN and the RBS (UTR1) and GFP sequence in a bicistronic design with the $cro$ gene (Supplementary Table 1)[30]. This plasmid served as the template for all further modifications that were introduced with PCR using phosphorylated primers, ligated (T4 Ligase, ThermoFisher Scientific), and directly transformed into bacteria. Constructs were sequenced by the sequencing service of the Weizmann Institute.

The $cI^{ts}$-*mvenus* construct was generated by fusing the C-terminus of the $cI^{ts}$ gene to the codon-optimized and truncated *dmvenus* sequence[30] with the point mutations for fast maturation and the monomeric state (L64V and A202K) resulting in *dmvenus-NB* (also known as SYFP2). Here for simplicity, we termed the fluorescent protein mVenus. The two proteins were genetically fused by a flexible linker (KRAPGTS, AAGCGAGCTCCCGGGACCAGC). The gene fragment of *mvenus* and linker was ordered as DNA fragment (gBlock, IDT) and after removing the *gfp* gene, cloned into the plasmids of the mono- and bistable GRNs using transfer-PCR.

**DNA protection.** Linear DNA is quickly degraded in cell-free extract without protection from enzymatic activity. RecBCD is essential for recombination in bacteria, but is known as key factor in the degradation of linear and single-stranded DNA. Commonly, the DNA-mimicking lambda bacteriophage protein GamS is supplemented in high amounts (~1 μM) to outcompete RecBCD[50]. But, to further improve the stability of DNA at the low-density regime, we devised a protocol to avoid using linear DNA, that is to prepare fluorescently labeled and biotinylated DNA without open ends at high concentrations for controlled surface immobilization.

The assembly protocol for the circular and hairpin DNA was the following:
Two PCR mixtures were prepared according to the manufacturer's protocol. The first mixture contained the template, one phosphorylated primer pP1, and one primer P2 (that is the sequence of pP2 without phosphorylation and overhang complementary to P1) with internal modification, e.g., biotinylated thymidine residue. The second PCR mixture contained the same template, a phosphorylated primer pP2 and a primer P1 (that is the sequence of pP1 without phosphorylation

and overhang complementary to P2) with internal modification, e.g., internal Cy5 label. After PCR purification, the products hold two identical dsDNA fragments except for the end modifications (P1, P2). For the final step, the two dsDNA fragments were mixed (2.5 µg of each in 50 µl reaction volume) and incubated with lambda exonuclease (5 units, NEB) in lambda exonuclease reaction buffer at 37 °C for 1 h to selectively degrade the end-phosphorylated (pP1, pP2) ssDNA parts (Supplementary Fig. 3). The reaction volume can be adjusted to prepare more DNA. The exonuclease was heat inactivated for 15 min at 75 °C. The product was eluted after clean-up from the spin column with ~25 µl buffer (10 mM Tris-HCl, pH 7.8, and 250 mM NaCl) and stored at −20 °C.

**Closed system gene expression**. Calibrations (Supplementary Fig. 6) and GRN control experiments (Supplementary Fig. 11) were done in a real-time PCR system (StepOnePlus and StepOnePlus v2.3, Applied Biosystems) that allowed simultaneous gene expression at six different temperatures. The system was calibrated with purified GFP in 1× phosphate-buffered saline (PBS) to linearize the readout. All solution experiments were conducted in a volume of 10 µl with 1 nM of plasmid. The conditions for cell-free gene expression were optimized to reduce temperature variations in basal GFP expression. For this batch of extract, a level of $Mg^{2+}$ and polyethylene glycol 8000 at 10 mM and 4%, respectively, gave ~10% expression differences in the range between 30 and 41 °C (Supplementary Fig. 1).

For the transient temperature variations in the solution experiments (closed system), we further implemented iLOV as fluorescent reporter that was shown to be oxygen-independent, temperature insensitive, and quickly maturated (Supplementary Fig. 11)[51]. However, we found that the brightness is not high enough to also use it as reporter for gene expression in the open systems.

At constant temperatures, all solution experiments were conducted with a plate reader (ClarioStar Plus, BMG Labtech) with 1 nM plasmid in a volume of 10 µl (Supplementary Figs. 3 and 17).

**Fabrication of microfluidic chips**. The mold for the microfluidic chip was fabricated using standard clean-room equipment and SU-8 photoresist (MicroChem, Newton, MA). A clean 4″ silicon wafer (4″, 0.525 mm thickness, <100>, p-type, University Wafers) was carefully covered with hexamethyldisilazane (HMDS, Transene Company) and incubated for 30 s to promote adhesion of the SU-8 photoresist. After incubation, the wafer was spun (PWM32, Headway Research Inc., Garland, TX) for 30 s at 3000 rpm with a ramp of 1000 rpm $s^{-1}$. The wafer was coated (compartment layer) with a 3 µm layer of SU-8 2002 (stage 1: 7.5 s at 500 rpm with acceleration of 100 rpm $s^{-1}$, stage 2: 30 s at 750 rpm with acceleration of 300 rpm $s^{-1}$). The photoresist was processed according to the manufacturer's protocol with a heating plate and exposed after careful alignment using 5″ chrome photomasks (Nanofilm) and mask aligner (Karl Suss Ma6/BA6, Garching, Germany). The patterned photoresist was treated according to the manufacturer's protocol with a heating plate omitting the hard baking step. The second layer (feeding layer) with 65 µm of SU-8 3050 was spun on the wafer (stage 1: 7.5 s at 500 rpm with acceleration of 100 rpm $s^{-1}$, stage 2: 30 s at 2000 rpm with acceleration of 300 rpm $s^{-1}$) and again processed according to the manufacturer's protocol with the final hard baking step (slow ramping from room temperature to 150 °C with slow cool-down to room temperature without removing the wafer from the heating plate).

The SU-8 mold with 9 microfluidic chips was covered with polydimethylsiloxane (PDMS, 10:1 ratio of polymer and curing agent, Sylgard 184, Dow Corning) in a petri dish, air bubbles were removed under vacuum for ~2 h, and baked overnight at 70 °C. The cured PDMS block was gently peeled off the wafer and cut into pieces. The holes for the inlet and outlet were punched on a cutting mat (0.75 mm diameter, Welltech Labs), thoroughly cleaned with isopropanol, and blow-dried with nitrogen. Fresh 170 µm coverslips (No. 1.5H, Marienfeld) were assembled in a Teflon holder, boiled in 96% ethanol for 10 min, and transferred to a clean beaker with 1:3 $NH_3(25\%):H_2O$ to be heated at 70 °C. Once heated, one part of $H_2O_2$ was supplemented and boiled for another 10 min. The coverslips were transferred into clean $H_2O$ and blow-dried with nitrogen for storage. Coverslips and PDMS parts were plasma treated at 35 watts for 30 s with 1.5 sccm $O_2$ inflow (Plasma System and GCM-200, March Plasmod) and brought in firm contact immediately after. The assembled chip was finalized by baking at 70 °C overnight.

**DNA patterning for low and high gene density**. An amount of 0.3 mg of biocompatible photoactivatable polymer solution (termed "DAISY")[52] was dissolved in 1.5 mL acetonitrile (HPLC graded) and flushed onto four chips through PTFE tubing (Cole Parmer). The microfluidic chips were incubated for 20 min to form a self-assembled monolayer on the surface. The solution was then successively exchanged by flushing 1 ml of 100, 50, 25, and 0% of acetonitrile (HPLC graded, Bio Lab LTD, Israel) and $H_2O$ mixture. The unprotected DAISY chains were blocked prior to UV illumination using Methyl-PEG$_4$-NHS (ThermoFisher Scientific, 5 mg ml$^{-1}$ in 250 mM borate buffer, pH 8.6) for high gene density. The chip was thoroughly washed with $H_2O$ after a 5-min incubation.

Photolithography was then performed in a UV cube (UV KUB, Kloé, France) for low gene density or on a standard optical microscope (Axiovert 200 M, Zeiss) with a mercury arc lamp (X-Cite Series 120Q, Excelitas Technologies Corp, USA)

for high gene density and spatial patterning. For low gene density, the chip was exposed continuously for 100 s at 100% power. For high gene density, the excitation spectrum was filtered using a 365 ± 10 nm band-pass filter (Chroma). A 200 µm spherical aperture (P200D, Thorlabs) was introduced as field stop into the microscope and a second adjustable spherical aperture was used to reduce the numerical aperture, hence reducing the influence of defocusing. The pattern was then passed through an 40×/0.75 NA objective (Olympus) and projected onto the DAISY treated microfluidic chip to expose (500 mJ cm$^{-2}$) a circular area of 14 µm diameter. A custom software written in Python 3.7 and PyQt5 using the µManager API (v1.4) allowed the automatic exposure of DAISY in each compartment. After exposure, the chip was incubated with Biotin-NHS (ThermoFisher Scientific, 5 mg ml$^{-1}$ in 250 mM borate buffer, pH 8.6) for 30 min, flushed with T250 buffer (10 mM Tris-HCl, pH 7.8, and 250 mM NaCl), and passivated with 0.1% Tween 20 in T250. Four chips were prepared in parallel and kept at room temperature (~20 °C) in a closed plastic box to prevent dehydration.

**Protein expression in the open system at the high gene density regime**. The patterned and functionalized microfluidic chips were flushed with biotinylated Cy5 labeled circular DNA at a concentration of 50–75 nM and 70–105 nM streptavidin (ratio of 1:1.4) in T250 with 100 nM dummy DNA (non-coding DNA) and incubated for 1 h at room temperature. Afterward, the chip was flushed with 2 ml of T250 buffer to remove any excess of DNA. The Cy5-DNA immobilization was verified by exciting the fluorophore with 12 mW at 647 nm (Colibri2 LED illumination, Zeiss), collecting the emission signal with an 40×/0.75 NA objective (Olympus) through a Cy5 filter (Chroma), and imaging the signal on an EM-CCD camera (500 ms, 250 gain, iXon 987, Andor Technology, Belfast, UK). Gene expression was performed on a standard optical microscope (Observer.Z1, Zeiss) with automated X, Y, and Z stage and auto-focus system (Zen 2012, Zeiss). The temperature was controlled using a top stage heating chamber (Boldline, Okolab); the humidity was passively controlled by introducing small water reservoirs into the heating chamber. Images were acquired in an interval of 1.5 min with the same hardware settings as used for the DNA signal with an LED excitation intensity of 29.5 mW at 488 nm (filter set 38 HE, Zeiss). The cell-free expression system was kept at 4 °C after preparation and introduced through a tubing (Tygon) using a 500 µL syringe pump (Gastight, Hamilton) with a constant negative flow rate of 0.4 µL min$^{-1}$ (PHD Ultra, Harvard Apparatus).

**Computing the GFP level in individual compartments**. To compute the GFP dynamics in the compartment, we outlined each compartment with a circular mask using Fiji (ImageJ v1.51p) and averaged the intensity. Since we found that PDMS fluoresced well within the GFP excitation spectrum and also bleached on a long timescale (~hours), we subtracted the background signals taken from the vicinity of each compartment at every time point. As a final step and to further improve the sensitivity of our system, we subtracted the auto-fluorescence signal of the cell-free extract as measured inside a compartment without DNA for every time point.

**Protein expression in the open system at the low gene density**. The functionalized microfluidic chips were flushed with biotinylated Cy5 labeled circular DNA at a concentration of 30 nM and 42 nM streptavidin (ratio of 1:1.4) in T250 and incubated for 30 min. The microfluidic chip was thoroughly washed with T250. Images were acquired on a custom-build single-molecule TIRF microscope. Illumination lasers with 488 nm (100 mW, OBIS, Coherent) and 647 nm (120 mW, OBIS, Coherent) were collinearly combined (DMSP605 and 5xBB1-E02, Thorlabs) through an objective mounted on a XYZ stage (MBT612D/M, Thorlabs) into a single-mode fiber (P5-460B-PCAPC-1, Thorlabs). The fiber output was coupled into a mirror collimator (RC08FC-P01, Thorlabs) to expand the laser diameter to 8 mm, guided through an achromatic lens ($f$ = 150 mm, AC254-150-A-ML, Thorlabs), re-directed by a mirror and filter cube (TRF59906, Chroma), and focused onto the back focal plane of the TIRF objective (60x, 1.49 NA, Nikon) in an angle to allow total internal reflection at the PDMS/glass and $H_2O$/glass interface. The cleaned emission signal was focused through a tube lens (TTL200, Thorlabs) onto an EM-CCD (iXon Ultra 888, Andor Technology, Belfast, UK). The two lasers were selected with the Arduino microcontroller (Arduino control software v1.2)[53] and synchronized with the camera's exposure output trigger signal using digital modulation option of the laser controllers. The sample could be translated along XY (Märzhäuser Wetzlar) and the objective was mounted on a piezo stage to focus along Z (400 µm Fast PIFOC, PI) with an in-house build Delrin adapter for thermal insulation. The objective was further enclosed with resistive heating foil (HT10K, Thorlabs) to adjust the temperature using a PID controller (TE-48-20, TE Technology).

The temperature was adjusted and equilibrated for at least 15 min before the sample was placed on the holder. The DNA molecules were localized with 7 W cm$^{-2}$ at 647 nm (200 ms exposure time, 250 Gain). The microfluidic chip with the immobilized DNA was flushed with 10 µL of cell-free extract using a 10 µL pipette tip. The pipette tip was kept inside the inlet to maintain flow by gravity inside the main channel resulting in the replenishment of cell extract and removal of proteins that were produced from DNA immobilized in the main channel. Single-molecule expression dynamics were imaged with 15 W cm$^{-2}$ at 488 nm (200 ms exposure time, 250 gain) in an interval of 1 s for a total acquisition time of 50–130 min (Andor Solis v1.3, Oxford Instruments).

We corrected the acquired movies for thermal drift with a cross-correlation algorithm (OpenCV v4.4) using the autofluorescent signal of the cell extract at 488 nm excitation. Then, individual compartments were outlined with the circular Hough transform (scikit-image). Inside the compartment boundaries, single molecules were localized with a local gradient algorithm (adapted from Picasso v0.3.0[54]) after correcting the uneven illumination of TIRF using a contrast limited adaptive histogram equalization (scikit-image v0.15.0). Intensity and background were extracted from the raw images. Trajectories were generated according to the following rules (trackpy v0.4.2): spots must not jump more than ~440 nm between consecutive frames (jump distance <2 pixel) and stayed there for at least 3 s with a memory of 1 sec to account for blinking events.

With the automated detection workflow, we estimated a false positive rate by counting the detected spots during the first 15 s period in which no cell-free expression system diffused into the compartment yet, hence no production of proteins occurred. With that, we could estimate a false positive rate of ~1 protein per 15 s.

**Calculation of autocorrelation functions of production rates**. The autocorrelation function is computed for each compartment from the production rate $x(t)$ after 20 min with the time-average production rate $\langle x \rangle$, variance $\sigma_x^2$ for a time delay $\tau$ as (Supplementary Figs. 12, 14, and 20):

$$A(\tau) = \frac{\langle (x(t) - \langle x \rangle)(x(t + \tau) - \langle x \rangle) \rangle}{\sigma_x^2} \quad (1)$$

Since individual compartments show high variability in the time-averaged production rates, we also computed the ensemble-averaged by averaging individual ACFs (Supplementary Figs. 18 and 20).

**Theoretical models of CI regulation**. An equilibrium model can be derived according to the mass-action law for simple

$$I + D \rightleftharpoons ID$$

and cooperative binding

$$I + D \rightleftharpoons ID$$

$$I + ID \rightleftharpoons I_2D$$

of inhibitor $I$ binding to its operator site $D$. They give an equilibrium dissociation constants according to $K_D = I^n D / I_n D$ with $n = 1,2,\ldots$. By assuming no self-inhibition (here only valid for short times), the amount of $I$ during protein expression is given by

$$I = k_{P,I} t \quad (2)$$

where $k_{P,I}$ (nM min$^{-1}$) is the production rate of the repressor in the cell extract. Together with the equilibrium equations, the amount of free DNA with $D = D_0 - ID$, and the reporter production with $dg/dt = k_{P,G} D$, with $k_{P,G}$ (min$^{-1}$) being the production rate of the reporter, the final form is

$$\frac{dg}{dt} = \frac{k_{P,G} D_0}{\frac{(k_{P,I}t)^n}{K_D} + 1} \quad (3)$$

Integration with non-cooperative binding ($n = 1$) from $g = 0$ to $G$ and $t = 0$ to $T$, gives the accumulation of reporter $G$ according to

$$G = \frac{k_{P,G} D_0 K_D}{k_{P,I}} \ln \frac{k_{P,I} T + K_D}{K_D} \quad (4)$$

with $K_D$ (nM) being the affinity constant.

Cooperative binding ($n = 2$) of inhibitor $I$ changes the course of the fluorescent reporter $G$ with

$$G = \frac{k_{P,G} D_0}{\sqrt{\frac{k_{P,I}^2}{K_D}}} \tan^{-1} \sqrt{\frac{k_{P,I}^2}{K_D}} T \quad (5)$$

where $K_D$ (nM$^2$) is the affinity constant. Comparing the experimental data with the derived models (compare with Supplementary Fig. 17), gives an estimate of around a 100-fold stronger affinity of CI to the operators for the production of $G$.

The stochastic simulations to estimate the $k_{on}$ rates for CI to the operator sites were conducted with COPASI 4.20 software using the stochastic integration by the direct method (Gillespie algorithm). The following equations were implemented inspired by the model from Isaacs et al.[24] with the parameters given in Supplementary Table 2.

Binding of CI$^{ts}$ ($P$) to the three operator sites on DNA $D$:

$$P + D \rightleftharpoons D_1, \; P + D_1 \rightleftharpoons D_2, \; P + D_2 \rightleftharpoons D_3$$

Transcription reactions with the basal ($TX_0$) and auto-activated ($TX_a$) states to produce mRNA $R$:

$$D \xrightarrow{TX_0} R, \; D_1 \xrightarrow{TX_a} D_1 + R, \; D_2 \xrightarrow{TX_a} D_2 + R$$

Translation reaction with translation rate $TL$:

$$R \xrightarrow{TL} R + P$$

The removal of mRNA and CI$^{ts}$ are given by the degradation reaction of mRNA ($R$) with rate $mdeg$, and the dilution and temperature-dependent inactivation reactions of CI$^{ts}$ ($P$) with the rates that are given in Supplementary Table 3:

$$R \xrightarrow{mdeg} \emptyset,$$

$$P \rightarrow \emptyset, \; D_1 \rightarrow D, \; D_2 \rightarrow D_1, \; D_3 \rightarrow D_2$$

The data plotted in Fig. 3d and Supplementary Fig. 16 is the number of proteins $P$ produced in an interval size of 15 s and a total simulation time of 60 min (ensemble-averaged production rates was also extracted after 20 min in steady-state). Simulations were run from 31 to 41 °C in 1 °C steps and each temperature was simulated for $N = 20$.

**Data analysis**. All experimental data were analyzed and visualized with Python 3.7 and the complementary packages (Seaborn v0.9, Matplotlib v3.3, NumPy v1.16, and Scipy v1.3).

**Reporting summary**. Further information on research design is available in the Nature Research Reporting Summary linked to this article.

## Data availability

The authors declare that the data supporting the findings of this study are available within the paper and its supplementary information files. Raw images are available from the corresponding author upon reasonable request. Source data are provided with this paper. Any other relevant data are available upon reasonable request. Source data are provided with this paper.

## Code availability

The data analysis in this study was performed with Python code, which will be made available from the corresponding author upon reasonable request. Source data are provided with this paper and stored on Zenodo (DOI: 10.5281/zenodo.4072036). Source data are provided with this paper.

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

## Acknowledgements

We thank M. Levy and O. Vonshak for fruitful discussions and help with experiments; L. Tunik for help in the fabrication facilities; N. Stern, M. Schwarz-Schilling, and T. Tlusty for helpful comments on the manuscript; and T. Tlusty for the brilliant support on all levels throughout the project. We thank D. Garenne for his participation in the preparation of the TXTL system used in this work. pGEX iLOV was a gift from John Christie (Addgene plasmid #26587). We acknowledge funding from the Israel Science Foundation (RBZ and SSD, grant no. 1870/15), the United States – Israel Binational Science Foundation (RBZ and VN grant no. 2014400), from the Human Frontier Science Program (VN, grand no. RGP0037/2015) and the Minerva Foundation (RBZ and SSD, grant no. 712274). We also thank EMBO for the financial support of F.G. through a long-term fellowship (ALTF 598-2017).

## Author contributions

F.G. designed the system, performed experiments, and analyzed the data. All authors discussed data and wrote the manuscript.

## Competing interests

The authors declare no competing interests.
