## [Peer Review File · Nature Communications]

Reviewers' Comments:

Reviewer #1:

Remarks to the Author:

Authors analyse the dynamics of a genetic switch in a cell-free setup, emphasising the impact of stochasticity by measuring a high-copy vs a low-copy switch. Since one of the molecules (of the two involved in the switch) is thermosensitive, authors focus on its control to showcase the different behavioural features of the switch. Stochasticity comes from having very low number of components in the low-copy version of the switch.

I think it is an interesting study. I particularly like switches and noise, so I enjoyed reading the manuscript. Having an isolated switch allows for assessing the impact of noise due to only molecular numbers in stability. I think this should be the focus of the study; however, I find the title (and the background narrative) a bit misleading. Here some suggestions that would improve the readability (for an audience like me, of course), or would require some clarifications, of the ms:

1. Why "decision-making"? I fail to see a decision between several possible outcomes based on some type of information (i.e., decision-making). I see a switch, which can be more or less noisy. To me, the story is about the switch and not about decision-making.
2. Why "synthetic cell"? I fail to see a synthetic cell. I see a cell-free system. Authors focus on the cell-free system they engineer so I think "synthetic cell" is confusing.
3. Why "the limits of biological computation"? I fail to see any discussion about (bio)computation in the text (e.g. complexity, computability, information, etc.). Results talk about bistability, switch dynamics and stochasticity. To me, the ms reads more like biophysics than biocomputation. And I don't understand "our results demonstrate biological computation outside the limits of living cells".
4. Why fuzzy computing? I fail to see any feature of fuzzy computation. Take fuzzy logic, for instance, as opposed to traditional binary logic: with reference points, thresholds and so on. I think the paper would need further clarifications on the fuzzy computing front in order to establish parallelisms between the switch dynamics and fuzzy computation.

Considering the above, I think there is a reference that would be of interest to the authors: "Construction of an in vitro bistable circuit from synthetic transcriptional switches" by Kim et al. (Mol. Sys. Bio. 2006), which built a memory by plugging two switches together. This (and other papers that followed) would be a good complement to this analysis of thermosensitive components and noise. More specific comments follow:

5. Why 20 copies for the low-copy regime? I mean, why not 1, or 100 or 1000?
6. In the high-copy scenario authors measure Pr activity (via GFP). But in the low-copy scenario authors measure Prm activity (via mVenus). Why not checking for the same activity in both scenarios?
7. Authors set diffusion at $D=30 \mu^2/s$. Do you have a reference for that?
8. Authors claim to build a GRN with memory. However, I don't see (may be my bad!) any graph/plot that showcase this memory effect. Only Fig 2b bottom has some hysteresis-like line, which would account for memory/stability after CI deactivation. Is there any other type of memory within this network? I think bistability needs two (bi) stable states, right?
9. Line 149. On a similar note than in point 6: can we compare noise in high-copy Prm versus noise in low-copy Pr? Also, does the low-copy scenario have some stable patters? Or is it just all random activity?
10. Line 203. I find it amazing authors demonstrate spatial coupling of synthesis and DNA-binding. Mirny's work on that matter is a reference (indeed, authors cite one of the papers). There is a recent modelling paper on this (Stoof et al, "A model for the spatiotemporal design of gene regulatory circuits" ACS Syn Bio, 2019) and I wonder if the 100-fold higher rate calculated by the authors is consistent with this (of course, cell-free vs in-vivo might change things a lot).
11. Line 564. Authors state that simulations were run from 31 to 41 C. How was this modelled?

Affecting the rate m_{deg} ?

12. Supp Fig 2b. Is there any correlation in the data? Looks randomly distributed to me.

13. Supp Fig 7a second column. I don't understand: if all lines start at 40, shouldn't they have the same initial performance? I must be missing something.

To sum up: very nice study. Congratulations. I just think the decision-making / biocomputation / fuzzy background confuses the message a bit. But I enjoyed the results of the cell-free switch and its dynamics. Lastly, I would like to see a discussion on the take-home lessons from this study e.g. the implications of noise, or temperature or binding dynamics, and their use in future work. There is a big amount of information in this study, and any reader would like to see a author's discussion on it with key concepts.

Reviewer #2:

Remarks to the Author:

The manuscript by Greiss et al. ("Decision-making in a synthetic cell: the limits of biological computation") devises a TXTL-based experimental set-up to investigate systems-level properties arising through the operation of gene regulatory networks (GRNs) in isolated 'synthetic cell' regimes. By immobilizing circularized DNAs on a surface in microfluidic chips, the authors could experimentally address several interesting questions about the intersections of thermodynamics, kinetics, GRNs and information processing/computation. In particular the manuscript analyzes the transition from large to small numbers of transcriptional effector molecules under constant dilution and measures apparent rate enhancements resulting from co-expressional localization. These are fundamental, long-standing questions and developing approaches to study them experimentally, as the authors have done here, is an important contribution. The manuscript may be suitable for publication once the following points are addressed.

Major:

1) It is unclear how much the particular choice of the experimental syncell volume impacts the results. Can the authors summarize what they understand about how these thresholds play out: some combination of binding constants, GRN dynamics and volume seem to be important, but it is unclear how to make sense of this. There are many observations throughout the manuscript that point in this direction, but some explicit analysis and discussion on the point would really help.

2) Throughout the paper the authors make a point of single molecule control, here they investigated low N and High N regimes of transcriptional effectors. While transcription from an individual promoter is controlled by a single effector, it wasn't entirely clear that these were single molecule effects. While they did claim the ability to observe individual fluorescent proteins, this is an indirect report of transcriptional state. mRNA degradation and dilution both exist in this system.

3) As currently presented, it seems difficult to definitively conclude that co-expressional localization is playing a major role in the low density regime. Co-expressional localization appears to call for a greater local concentration of both DNA and CI than what the manuscript estimates (without affecting the binding rate). Moreover, the parameter values used in the estimates and the model were taken from characterizations done under very different conditions, usually at 22C or 28C. Overestimation of other parameters such as rate of unbinding and mRNA degradation rates could lead to the same conclusion. The manuscript would be improved by investigating the effect of overestimating these parameters, e.g. by varying them alongside the binding rate in the stochastic simulations.

4) How does co-expressional localization interact with the way that constant protein turnover occurs in this system? The experimental system requires constant protein turnover but the mechanism suggested for 100x higher K_{on} is co-expressional localization in an inhomogeneous

TXTL mixture (with diffusion partially constrained by compartmentalization). For instance, previous work has shown that PEG can increase the local concentration of substrates by partitioning them into smaller compartments, the length scale of the experimental setup is large compared to the size of compartment that you can expect from partitioning due to PEG. The authors should address the fact the TXTL mixture is inhomogeneous and decidedly not well mixed re: the constant turnover assumption.

5) The function of the GRN is being changed by both copy number and temperature. When making comparisons across different temperatures, what affect does temperature have on the diffusion coefficient? And, how does this interact with the observation of co-expressional localization (presumably due to partitioning within the PEG). The concern is that the effects of changing temperature in an inhomogeneous non- equilibrium mixture may not be obvious.

6) It is not clear why two different approaches were taken to probe promoter state at high- and low-density regimes. The authors should assess the activity of CI after tagging the C-terminus with dmvenus-NB to ensure that this is not skewing the low density results.

7) The manuscript explains that "Fuzzy logic" appeared at concentrations 100 fold lower than equilibrium -- where is that comparison and data in the paper? The authors should define the term more explicitly and point exactly to the places where the data show the highlighted phenomenon.

8) The manuscript ends rather abruptly and would be strengthened by more synthesis of the results and explaining what the results mean both for biological computing and for synthetic cell construction. For instance, the manuscript briefly mentions the idea of metabolic load as the source of a thermodynamic limit on computation. Although not the primary focus of the current manuscript, it would be interesting for the authors to explain what the current work says about where those limits are and how metabolic load is likely to scale with different computational operations -- for example, fuzzy logic.

Minor:

1) Not sure that TXTL is required to investigate these questions, but it is a suitable environment for those reasons. That sentence should be toned down.

2) In general, there are lots of specialist vocabulary that should be more clearly defined. For instance, explaining the Fano factor and the appropriate application should be described for a generalist journal (e.g. line 205).

3) What is the red line in Figure 2?

4) It would be helpful to add false positive rates of mVenus detection.

5) Line 87 "noiseless" would be better described as 'high signal to noise'.

6) Line 241 "devoid of any extrinsic noise" is confusing as written. Is this meant to be analogous to inter cellular variation replicate experiments with the same GRN circuit topology and reaction conditions? As in #2, this term should be more carefully explained.

Reviewer #1 (Remarks to the Author):

Authors analyse the dynamics of a genetic switch in a cell-free setup, emphasising the impact of stochasticity by measuring a high-copy vs a low-copy switch. Since one of the molecules (of the two involved in the switch) is thermosensitive, authors focus on its control to showcase the different behavioural features of the switch. Stochasticity comes from having very low number of components in the low-copy version of the switch.

I think it is an interesting study. I particularly like switches and noise, so I enjoyed reading the manuscript. Having an isolated switch allows for assessing the impact of noise due to only molecular numbers in stability. I think this should be the focus of the study; however, I find the title (and the background narrative) a bit misleading. Here some suggestions that would improve the readability (for an audience like me, of course), or would require some clarifications, of the ms:

We thank reviewer #1 for his/her interest in our study.

1. Why “decision-making”? I fail to see a decision between several possible outcomes based on some type of information (i.e., decision-making). I see a switch, which can be more or less noisy. To me, the story is about the switch and not about decision-making.

We agree with reviewer #1 that decision-making generates several possible outcomes based on information. Specifically for the lambda bacteriophage which is a classical system for decision-making, the decision is made between lysis or lysogeny of the host cell by the viral genetic circuit. To this end, the genetic circuit takes the level of activated RecA as native input to the genetic decision-making. The level of active RecA depends on DNA damage in the host cell and can degrade CI through cleavage. If the level of RecA is high (high DNA damage, fast CI cleavage), the host cell will be lysed; if the level of RecA is low (no DNA damage, no CI cleavage), the host cell will continue growing with the viral genetic circuit (see “A genetic switch” by Ptashne, Ref. 16). As the lambda bacteriophage decides between those two outcomes (lysis/lysogeny) with information from DNA damage and maintains the outcome for many generations, we refer here to “decision-making”.

Here, we specifically replaced the information of DNA damage (and CI cleavage by RecA) by a well-studied temperature-sensitive mutation of CI (ci857) and used temperature as input to the “decision-making” process. As we increased the deactivation rate of CI^{ts} with temperature, the artificial cell toggles between the two promoters Pr and P_{rm} and can maintain this decision.

We were also encouraged to use the concept of “decision-making” by *in vivo* studies of the lambda bacteriophage genetic circuit. Please see a few exemplary citations from various labs (also cited in our manuscript):

- “The life cycle of bacteriophage lambda serves as a simplified paradigm for cell-fate decisions.” (see Golding, Ref. 17)
- “These findings uncover cell fate potentials beyond the classical picture of λ switch, and open a new window to explore the genetic and environmental origins of the cell fate decision-making process in gene regulatory networks.” (see Fang *et al.*, Ref. 18)
- “The lysis/lysogeny decision in phage λ is the classic example of a bistable switch, with switching mediated by a double negative feedback loop between repressors CI and Cro” (see Norman *et al.*, Ref. 19)
- “This is a typical example of decision making at the subcellular level, as viruses with identical genomes infecting isogenic cells can either become lytic or lysogenic. Despite the apparent simplicity of the viral genome, the story of lambda phage decision making is still not completely written and may hold many surprises.” (see Balázsi *et al.*, Ref. 20)

2. Why “synthetic cell”? I fail to see a synthetic cell. I see a cell-free system. Authors focus on the cell-free system they engineer so I think “synthetic cell” is confusing.

We are building on the concept of bottom-up biology that reconstructs aspects of a living cell. Here, we reconstruct the aspect of gene expression (transcription and translation) from immobilized DNA with constant molecule turnover. The artificial cell gives a reaction volume that permits steady-state protein levels, also referred to homeostatic conditions, namely the production and dilution/degradation of molecules, hence a constant turnover of molecules inside a compartment. With that in mind, we refer to our published work on gene expression dynamics in artificial cells as defined by the compartment geometry in the microfluidic chip:

- Tayar, A. M., Karzbrun, E., Noireaux, V., & Bar-Ziv, R. H. (2015). Propagating gene expression fronts in a one-dimensional coupled system of artificial cells. *Nature Physics*, 11(12), 1037–1041. <http://doi.org/10.1038/nphys3469>
- Karzbrun, E., Tayar, A. M., Noireaux, V., & Bar-Ziv, R. H. (2014). Programmable on-chip DNA compartments as artificial cells. *Science*, 345(6198), 829–832. <http://doi.org/10.1126/science.1255550>
- Tayar, A. M., Karzbrun, E., Noireaux, V. & Bar-Ziv, R. H. Synchrony and pattern formation of coupled genetic oscillators on a chip of artificial cells. *Proc. Natl. Acad. Sci. U. S. A.* (2017) doi:10.1073/pnas.1710620114.

We added the references for clarification (Ref. 13-15) and renamed “synthetic cells” to “artificial cells”.

3. Why “the limits of biological computation”? I fail to see any discussion about (bio)computation in the text (e.g. complexity, computability, information, etc.). Results talk about bistability, switch dynamics and stochasticity. To me, the ms reads more like biophysics than biocomputation. And I don’t understand “our results demonstrate biological computation outside the limits of living cells”.

Decision-making is the simplest form of computation as defined by computability theory and computational complexity theory. We agree that it might confuse and hence removed the word “computation” from the manuscript. We changed the title, revised the abstract and removed the sentence: “our results demonstrate biological computation outside the limits of living cells”.

Given the functionality of decision-making of the lambda bacteriophage, the genetic circuit is capable to compute an output (promoter activity) from a given input (e.g. temperature). The decision-making between the two promoters gave a deterministic output (binary) in the high-density regime, whereas in the low-density regime the promoters could not be clearly assigned (fuzzy) for temperatures between 35-39C.

We observed the gene expression dynamics at very different gene densities (here referred as limits), namely a regime with low densities of genetic circuits and strong fluctuations and a regime with large densities of genetic circuits and low fluctuations. These limits usually could not be obtained in living cells due to factors, such as high metabolic load on living bacterial cells with the strong gene expression of auxiliary gene circuits and a limit on the amount of auxiliary DNA (e.g. plasmids) that can be maintained in a single bacterial cell. As the dynamics of genetic circuits in living cells are also intertwined with many other processes, i.e. resource sharing of ribosomes, tRNA, ..., we wanted to highlight that our work opens a way to look at the isolated dynamics of a bistable circuit.

4. Why fuzzy computing? I fail to see any feature of fuzzy computation. Take fuzzy logic, for instance, as opposed to traditional binary logic: with reference points, thresholds and so on. I think the paper would need further clarifications on the fuzzy computing front in order to establish parallelisms between the switch dynamics and fuzzy computation.

We agree with reviewer #1 that the term “fuzzy computation” is not well defined. We clarified the main text and replaced “fuzzy computation” by “fuzzy decision-making”.

Considering the above, I think there is a reference that would be of interest to the authors: “Construction of an in vitro bistable circuit from synthetic transcriptional switches” by Kim et al. (Mol. Sys. Bio. 2006), which built a memory by plugging two switches together. This (and other papers that followed) would be a good complement to this analysis of thermosensitive components and noise. More specific comments follow:

We thank reviewer #1 for the inspiring reference. The publication “Construction of an *in vitro* bistable circuit from synthetic transcriptional switches” by the Winfree lab gives a nice example of a synthetic transcriptional bistable switch lacking the production and degradation of proteins.

We gladly added a comment and referenced the publication (Ref. 46).

5. Why 20 copies for the low-copy regime? I mean, why not 1, or 100 or 1000?

As we were interested in the properties of the genetic switch with the lowest number of gene circuits possible in our setup, we chose to study the dynamics of gene expression with a typical number of 20 copies. Here, we found that the stochastic DNA immobilization was well-behaved and reproducible. The ~20 DNA molecules are however an average from a distribution that we report in Supplementary Fig. 2a. We agree with the reviewer that the text was not clear.

We now write “We could not further reduce the number of DNA molecules while maintaining a narrow distribution due to the diffusion-based stochastic immobilization in our artificial cells.”

6. In the high-copy scenario authors measure Pr activity (via GFP). But in the low-copy scenario authors measure Prm activity (via mVenus). Why not checking for the same activity in both scenarios?

We found that the elastomer (PDMS) is highly fluorescing (~488 nm excitation) with epifluorescence microscopy. Using the strong P_R promoter, we could increase the fluorescence output of the genetic circuit above the fluorescence background of PDMS. We observed gene expression dynamics with very high signal to noise ratio and no deviations between compartments at the same temperature. As the typical time-scale of gene expression dynamics in the high-density regime were ~1 h, the longer maturation time of GFP compared to mVenus-NB should not interfere with our conclusions.

Next, we performed the single-molecule experiments in the low-density regime with mVenus-NB and its improved photo-physical properties (Ext. coefficient: 101,000, QY: 0.68, Brightness: 68.68) compared to GFP (Ext. coefficient: 55,900 $M^{-1}cm^{-1}$, QY: 0.6, Brightness: 33.54). Here, we used the well-characterized approach of fusing the C-terminus of CI to the fluorescence reporter (see Rosenfeld *et al.*, Ref. 11). The fusion enabled us to measure the precise 1:1 stoichiometry and to allowed us to obtain the quantitative information needed for the Fano factor and ACF analysis.

To avoid confusion of the reader we changed the text accordingly. We added a statement for the high-density regime: “The strong P_R promoter increased the GFP signal over the background fluorescence of the elastomer used to create the compartments. We observed smooth GFP dynamics with a high signal to noise ratio, ...”

We also added a statement for the low-density regime: “To allow the precise monitoring of protein production, we directly fused the cI^{ts} gene to the fast-maturing fluorescent protein mVenus under the weak P_{RM} promoter, ...”

7. Authors set diffusion at $D=30 \mu\text{m}^2/\text{s}$. Do you have a reference for that?

Thank you for pointing out the missing reference. In earlier work, we measured the diffusion coefficient in the TXTL mixture (see Karzbrun *et al.*, Ref. 13). We added the reference at the appropriate position.

Other reported values for the diffusion coefficient of GFP range from 10-90 $\mu\text{m}^2/\text{sec}$ (<https://bionumbers.hms.harvard.edu/bionumber.aspx?s=n&v=3&id=112266>).

8. Authors claim to build a GRN with memory. However, I don't see (may be my bad!) any graph/plot that showcase this memory effect. Only Fig 2b bottom has some hysteresis-like line, which would account for memory/stability after CI deactivation. Is there any other type of memory within this network? I think bistability needs two (bi) stable states, right?

As the reviewer correctly pointed out, Fig2b shows memory after CI deactivation. The landmark studies of the Ptashne lab and others made clear that with the genetic switch of the lambda bacteriophage two stable states can exist: one being inactive Pr / active P_{RM} and the other being active Pr / inactive P_{RM}. With this said, we do not claim that we build a bistable GRN, but isolated the core element of a known and well-studied natural bistable GRN with memory and incorporated it into an artificial environment.

Since the two repressors CI and Cro inhibit each other, they form a mutually repressing genetic topology. Specifically, when CI is produced, CI continues to repress Cro production; when Cro is produced, Cro continues to repress CI. The temperature-sensitive mutation of CI^{ts} allows to switch from active CI to active Cro with an increase in temperature.

9. Line 149. On a similar note than in point 6: can we compare noise in high-copy P_{RM} versus noise in low-copy Pr? Also, does the low-copy scenario have some stable patterns? Or is it just all random activity?

We do not quantitatively compare the noise between the low- and high-density regime in the framework of Fano factor and ACF. As stated in the main text and unlike the low-density regime, we could not observe any difference between compartments at the same temperature in the high-density regime. We found that dynamics in the high-density regime are essentially smooth, deterministic, and noise-free. Noise in gene expression only appeared at the low-density regime.

Other publications such as Raj and van Oudenaarden (Ref. 26) also pointed out that measuring noise without the resolution of single molecules can lead to

misinterpretations of the underlying genetic expression noise. Therefore, we only studied the fluctuations in gene expression in the low-copy regime with single-molecule resolution using the Fano factor and ACF. With those measures, we were able to show that fluctuations (random activity) are greatly reduced for the low-density regime outside the switching temperature of 35-39C.

10. Line 203. I find it amazing authors demonstrate spatial coupling of synthesis and DNA-binding. Mirny's work on that matter is a reference (indeed, authors cite one of the papers). There is a recent modelling paper on this (Stoof et al, "A model for the spatiotemporal design of gene regulatory circuits" ACS Syn Bio, 2019) and I wonder if the 100-fold higher rate calculated by the authors is consistent with this (of course, cell-free vs in-vivo might change things a lot).

We thank the reviewer for pointing out the reference from the Goñi-Moreno lab and agree that direct comparison of cell-free experiments with *in vivo* experiments could be difficult. After carefully reading the publication, we could not find a direct reference towards co-expressional localization, but 1-D sliding and 3-D diffusion of transcription factors after translation. This is now referenced in the revised manuscript (Ref. 37).

We included a reference to earlier work from Riggs *et al.* (1970) and others (Ref. 36, 38, 39) that discussed a 100-1000 times higher k_{on} rates for the Lac repressor to DNA and discussed the model of facilitated diffusion by 1-D sliding and 3-D diffusion. However, we want to stress that we did not measure the k_{on} rate experimentally, but inferred an order-of-magnitude estimate from simulations and solution experiments. We want to take the chance and mention the alternative models in the discussion of our manuscript.

11. Line 564. Authors state that simulations were run from 31 to 41 C. How was this modelled? Affecting the rate mdeg?

As pointed out in the manuscript, the temperature and with that the inactivation of Cl^{ts} was modeled with reported inactivation rates of Cl^{ts} . Please see the Methods section, Supplementary Table 3, and the publication by Isaacs et al. (Ref. 24). To further clarify, we modified the corresponding text in the theoretical model section.

12. Supp Fig 2b. Is there any correlation in the data? Looks randomly distributed to me.

We agree with the reviewers comment that the data in Supplementary Fig. 2b looks randomly distributed as we also pointed out in the main text "We also found no dependence of protein production rates on the slightly varying number of DNA molecules in the low-density compartments".

13. Supp Fig 7a second column. I don't understand: if all lines start at 40, shouldn't they have the same initial performance? I must be missing something.

As correctly pointed out by the reviewer #1, the initial temperature was set to 40C and reduced to the indicated temperatures after 1 h (also as stated in the main text). We agree with the comment that the initial dynamics vary slightly (<2-fold) between the temperature conditions. Slight variations due to the assembly of the microfluidic chips might account for the different responses.

We added the following text for clarification: "The slight variation in the initial response (<1h) is explained by the small variations of the microfluidic chips during the fabrication process."

To sum up: very nice study. Congratulations. I just think the decision-making / biocomputation / fuzzy background confuses the message a bit. But I enjoyed the results of the cell-free switch and its dynamics. Lastly, I would like to see a discussion on the take-home lessons from this study e.g. the implications of noise, or temperature or binding dynamics, and their use in future work. There is a big amount of information in this study, and any reader would like to see a author's discussion on it with key concepts.

We thank the reviewer #1 for his insights and comments. We added an extended discussion to the main text.

Reviewer #2 (Remarks to the Author):

The manuscript by Greiss et al. ("Decision-making in a synthetic cell: the limits of biological computation") devises a TXTL-based experimental set-up to investigate systems-level properties arising through the operation of gene regulatory networks (GRNs) in isolated 'synthetic cell' regimes. By immobilizing circularized DNAs on a surface in microfluidic chips, the authors could experimentally address several interesting questions about the intersections of thermodynamics, kinetics, GRNs and information processing/computation. In particular the manuscript analyzes the transition from large to small numbers of transcriptional effector molecules under constant dilution and measures apparent rate enhancements resulting from co-expressional localization. These are fundamental, long-standing questions and developing approaches to study them experimentally, as the authors have done here, is an important contribution. The manuscript may be suitable for publication once the following points are addressed.

We thank reviewer #2 for his/her interest in our study.

Major:

1) It is unclear how much the particular choice of the experimental syncell volume impacts the results. Can the authors summarize what they understand about how these thresholds play out: some combination of binding constants, GRN dynamics and volume seem to be important, but it is unclear how to make sense of this. There are many observations throughout the manuscript that point in this direction, but some explicit analysis and discussion on the point would really help.

We thank the reviewer for pointing out the compartment volume as factor in gene regulation. We chose the reported compartment geometry to reach a protein life-time in the range of many minutes as we also reported in our recent publications on artificial cells. We did not change the volume of the cell, but only the compartment life-time by the length of the capillary in the high-density regime and left it constant in the low-density regime, since we could not observe an effect on the switching temperature. Currently, it is challenging to enter the scale of bacterial volumes, since the protein life-time scales with the geometry and would enter sub-minutes at the scale of a bacterial cell. It would be very interesting to further reduce the volume down to a single bacteria, but this is currently very challenging and is left for future work.

We were surprised that the concentration seemed to have little influence on the dynamics of the genetic circuit as the switching temperature did not change for high-density and low-density regime. We also found a similar temperature response in solution experiments without compartmentalization (see Supplementary Fig. 11). We could only understand the observed regulation by co-expressional localization.

As we pointed out in question 10 of reviewer #1, we gladly take the chance and discuss the current models of gene regulation in bacteria. We added a discussion to the manuscript.

2) Throughout the paper the authors make a point of single molecule control, here they investigated low N and High N regimes of transcriptional effectors. While transcription from an individual promoter is controlled by a single effector, It wasn't entirely clear that these were single molecule effects. While they did claim the ability to observe individual fluorescent proteins, this is an indirect report of transcriptional state. mRNA degradation and dilution both exist in this system.

Our measurements in the low-density regime were performed with single-molecule resolution, but the effect of gene expression dynamics are integrated over the entire compartment. Therefore, we monitored the protein production from an average of 20 DNA molecules. We agree with reviewer #2 that single-molecule gene regulation might confuse the reader and changed the sentence to "gene regulation with a few molecules" in our manuscript.

We agree with the reviewers comment that the production and degradation or dilution of mRNA as well as the production and dilution of proteins play a major role

in our compartments. Up to this day, it has been challenging to simultaneously observe translation and transcription dynamics on the single-molecule level due to technical limitations. One outstanding exception is the study by Sepúlveda *et al.* (Ref. 9) that relied on the fixation of bacterial cells to obtain single snapshots of transcription and translation and inferred the gene expression dynamics from theoretical considerations.

3) As currently presented, it seems difficult to definitively conclude that co-expressional localization is playing a major role in the low density regime. Co-expressional localization appears to call for a greater local concentration of both DNA and CI than what the manuscript estimates (without affecting the binding rate). Moreover, the parameter values used in the estimates and the model were taken from characterizations done under very different conditions, usually at 22C or 28C. Overestimation of other parameters such as rate of unbinding and mRNA degradation rates could lead to the same conclusion. The manuscript would be improved by investigating the effect of overestimating these parameters, e.g. by varying them alongside the binding rate in the stochastic simulations.

We performed more stochastic simulations to investigate the other parameter effects (mRNA half-life and k_{off} rates) on the properties of protein production rates with temperature (see Fig. 2). Unlike the scan in k_{on} rates for the repressor CI to its operator sites that reproduced the experimental data as we initially reported in our manuscript, we could not match the production rates over temperature with a scan in the other parameters. These results strengthen our conclusion that k_{off} and mRNA degradation rates are less important compared to the CI's k_{on} rate for gene regulation in the artificial cells.

We included the information in the supplementary information as Supplementary Fig. 16.

Figure 1: Stochastic simulation of monostable GRN with various parameter scans. **a)** Scan of k_{off} rates of CI (0.25 min^{-1} was used in the main text as described in the methods part and the supplementary information). The simulated shape of the CI self-regulation showed no major effect from a scan in k_{off} rate. **b)** Scan of mRNA degradation rates (0.5 min^{-1} was used in the main text as described in the methods part and the supplementary information). The simulated shape of the CI self-regulation showed no major effect from a scan in mRNA degradation rate. As expected, the amplitude of protein production rates increased with longer mRNA half-life.

4) How does co-expressional localization interact with the way that constant protein turnover occurs in this system? The experimental system requires constant protein turnover but the mechanism suggested for 100x higher K_{on} is co-expressional localization in an inhomogenous TXTL mixture (with diffusion partially constrained by

compartmentalization). For instance, previous work has shown that PEG can increase the local concentration of substrates by partitioning them into smaller compartments, the length scale of the experimental setup is large compared to the size of compartment that you can expect from partitioning due to PEG. The authors should address the fact the TXTL mixture is inhomogeneous and decidedly not well mixed re: the constant turnover assumption.

We thank the reviewer for his insights on partitioning in the TXTL mixture by the crowding agent PEG. Looking at the fluorescence signal in the low- and high-density regime, we fail to detect an inhomogeneous distribution or clustering that would be expected from a slow dispersion of molecules. In case that partitioning would happen in the cell-free expression system and would tune the local concentration/diffusion, CI^{ts} would still be deactivated by temperature or would leave the operator site after ~4 minutes (see k_{off} rate in the Supplementary Table 3) to free the binding site for other newly synthesized CI proteins, hence a constant turn-over of CI.

We included the publication by the Huck lab (see Ref. 41 and 42) that reported on the partitioning of the cell-free expression system by crowding agents and their effects on gene expression.

We also added a discussion on the possibility of TXTL partitioning.

5) The function of the GRN is being changed by both copy number and temperature. When making comparisons across different temperatures, what affect does temperature have on the diffusion coefficient? And, how does this interact with the observation of co-expressional localization (presumably due to partitioning within the PEG). The concern is that the effects of changing temperature in an inhomogeneous non- equilibrium mixture may not be obvious.

We agree with the comment of reviewer #2 that temperature effects on a non-equilibrium mixture is difficult to predict. Therefore, as experimental control for these temperature effects, we included the genetic circuit lacking the temperature-sensitive mutation of CI to show that temperature effects are minor compared to the regulatory feedbacks and deactivation of CI^{ts} (see Supplementary Fig. 14 and Supplementary Table 2).

We added a statement to clarify the role of the control experiment with wild-type CI towards diffusion coefficient and non-equilibrium partitioning by PEG:

“A control experiment of a monostable GRN with the wild-type *cI* gene displayed almost no temperature response, consistent with the temperature having no effects on the GRN other than the deactivation of CI^{ts} -mVenus.”

6) It is not clear why two different approaches were taken to probe promoter state at high- and low-density regimes. The authors should assess the activity of CI after

tagging the C-terminus with dmvenus-NB to ensure that this is not skewing the low density results.

We followed a well-characterized approach by Rosenfeld *et al.* (Ref. 11) of fusing the C-terminus of CI to the fluorescent reporter. Please also see question 6 of reviewer #1.

7) The manuscript explains that "Fuzzy logic" appeared at concentrations 100 fold lower than equilibrium -- where is that comparison and data in the paper? The authors should define the term more explicitly and point exactly to the places where the data show the highlighted phenomenon.

We thank reviewer #2 for his/her comment. We rewrote the abstract stating "Fuzzy computation appeared at DNA and protein concentrations 100-fold lower than necessary in equilibrium, suggesting rate enhancement by co-expressional localization." to "Gene regulation was observed at lower DNA and protein concentrations than necessary in equilibrium, suggesting rate enhancement by co-expressional localization."

8) The manuscript ends rather abruptly and would be strengthened by more synthesis of the results and explaining what the results mean both for biological computing and for synthetic cell construction. For instance, the manuscript briefly mentions the idea of metabolic load as the source of a thermodynamic limit on computation. Although not the primary focus of the current manuscript, it would be interesting for the authors to explain what the current work says about where those limits are and how metabolic load is likely to scale with different computational operations -- for example, fuzzy logic.

We added a discussion to the manuscript.

Minor:

1) Not sure that TXTL is required to investigate these questions, but it is a suitable environment for those reasons. That sentence should be toned down.

We agree with reviewer #2. We now write: "In order to address these questions experimentally without background reactions such as DNA replication, genetic cross-talk, resource sharing, or cell size variations, we used a programmable cell-free system of gene-expression reactions with constant protein turnover."

2) In general, there are lots of specialist vocabulary that should be more clearly defined. For instance, explaining the Fano factor and the appropriate application should be described for a generalist journal (e.g. line 205).

Thank you for pointing this out. We changed the manuscript accordingly.

3) What is the red line in Figure 2?

Thank you for pointing this out. The line should guide the eye. We now write in the figure caption “Red line drawn to guide the eye”.

4) It would be helpful to add false positive rates of mVenus detection.

We applied the same detection algorithm throughout the entire set of experiments with different conditions and genetic topologies. Therefore, we detect the same number of false positives throughout the experiments and do not compromise the comparison of the noise analysis in gene expression at the low-density regime.

By counting the fluorescent spots that were detected before the cell-free expression system fully enters the compartment, we can estimate a false positive rate of ~1 per 15 seconds. We added the discussion to the methods sections for further clarification.

5) Line 87 “noiseless” would be better described as ‘high signal to noise’.

We agree with reviewer #2 and now write “We observed smooth GFP dynamics with a high signal to noise ratio, ...”

6) Line 241 “devoid of any extrinsic noise” is confusing as written. Is this meant to be analogous to inter cellular variation replicate experiments with the same GRN circuit topology and reaction conditions? As in #2, this term should be more carefully explained.

Thank you for pointing this out. In the main text, we introduced extrinsic noise as DNA replication, genetic cross-talk, and cell size variations: “In order to address these questions experimentally without background reactions such as DNA replication, genetic cross-talk, resource sharing, or cell size variations, we used a programmable cell-free system of gene-expression reactions with constant protein turnover.”

We changed the text and now write “...devoid of any background reaction in living cells.”

Reviewers' Comments:

Reviewer #1:

Remarks to the Author:

I think the revised manuscript highlights the results in a better way. Authors addressed all points, in what has been a very nice (kind of) discussion. I encourage the journal to publish this manuscript as is.

Reviewer #2:

Remarks to the Author:

I have reviewed the rebuttal file and the revised manuscript. The authors have addressed all of my concerns.

Reviewer #1 (Remarks to the Author):

I think the revised manuscript highlights the results in a better way. Authors addressed all points, in what has been a very nice (kind of) discussion. I encourage the journal to publish this manuscript as is.

Reviewer #2 (Remarks to the Author):

I have reviewed the rebuttal file and the revised manuscript. The authors have addressed all of my concerns.